# A test of stochastic parroting in a generalisation task: predicting the characters in TV series

## Abstract

There are two broad, opposing views of the recent developments in large language models (LLMs). The first of these uses the term "stochastic parrots" from Emily Bender et al [3] to emphasise that because LLMs are simply a method for creating a probability distribution over sequences of words, they can be viewed as simply parroting information in the training data. The second view, "Sparks of AGI" from Sebastien Bubeck et al [6], posits that the unprecedented scale of computation in the newest generation of LLMs is leading to what its proponents call "an early (yet still incomplete) version of an artificial general intelligence (AGI) system". In this article, we propose a method for making predictions purely from the representation of data inside the LLM. Specifically, we create a logistic regression model, using the principal components of a LLM model embedding as features, in order to predict an output variable. The task we use to illustrate our method is predicting the characters in TV series, based on their lines in the show. We show that our method can, for example, distinguish Penny and Sheldon in the Big Bang Theory with an AUC performance of 0.79. Logistic regression models for other characters in Big Bang Theory have lower values of AUC (ranging from 0.59 to 0.79), with the most significant distinguishing factors between characters relating to the number and nature of comments they make about women. The characters in the TV-series Friends are more difficult to distinguish using this method (AUCs range from 0.61 to 0.66). We find that the accuracy of our logistic regression on a linear feature space is slightly lower than GPT-4, which is in turn at a level comparable to two human experts. We discuss how the method we propose could be used to help researchers be more specific in the claims they make about large language models.

## 1 Introduction

Large language models (LLMs) are neural networks trained on a large text corpus to predict the next word, phrase or paragraph in that dataset [25]. As the number of network parameters and the size of the corpus increases, the ability of this network to write convincing-sounding texts improves [15]. As a result, an increasing number of compelling LLM applications, from CHAT-GPT to Copilot, have been developed. Recently, Bubek et al. argued that "beyond its mastery of language, GPT-4 can solve novel and difficult tasks that span mathematics, coding, vision, medicine, law, psychology and more, without needing any special prompting" [6]. For these authors, this ability to generalise revealed "Sparks of AGI", going on to state that they believed "that [GPT-4] could reasonably be viewed as an early (yet still incomplete) version of an artificial general intelligence (AGI) system."

The stochastic parrots paradigm critiques such claims by pointing out that large language models simply predict the next word, sentence or paragraph, and it is humans who attribute understanding to its output [3]. LLMs simply replicate examples (i.e. parrot text) from a massive corpus of data [7].

The stochastic parrots view provides an epistemic critique of claims, such as "Sparks of AGI", about artificial general intelligence. For example, in the context of the benchmark tests (such as those later carried out by [6]), Raj et al. (2021) write, "the reality of [benchmark] development, use and adoption indicates a *construct validity* issue, where the involved benchmarks — due to their instantiation in particular data, metrics and practice — cannot possibly capture anything representative of the claims to general applicability being made about them." In other words, the very notion of generality, sought to be proven in "Sparks of AGI", cannot be captured by benchmark problems [6]. This critique is fundamental: it doesn't matter how many specific tasks a model completes, there is no convergence towards generality. Even setting these epistemic problems aside, the stochastic parrots view also has practical implications for how we evaluate LLM performance. For example, Lewis and Mitchell (2024) manipulate benchmark tasks to construct 'counterfactual' tasks, by for example adding information that solves the task but LLM's neglect this information, because they are parroting answers to similar, previously trained-on examples [18].

In spite of the limitation of benchmarks, the fact remains that LLMs do perform well over a wide range of tasks, with little or no additional training data. It is the question of understanding how such performance might arise which we address in this paper. Instead of proposing new benchmarks, we focus on comparing how LLMs perform to simpler, well-understood statistical methods on a novel task. An approach like ours has previously been persued medical imaging — where a systematic review showed that logistic regression on selected features performed (on average) just as well as complicated machine learning approaches [8] — and with respect to conflict prediction — logistic regression perform just as well (as is easier to interpret) than more complex machine learning models [16].

For many general tasks, a relatively straightforward method of making predictions is to use linear or logistic regression on the leading principal components of a data set. One example is using principle components of 'likes' of Facebook users to predict the answers people gave to big-five personality tests [32, 19, 17]. Konsinski et al. (2016) first performed PCA or Latent Dirichlet Allocation (LDA) on the matrix of likes and Facebook users, and then used the leading components of the PCA (or clusters of LDA) in a regression model to predict the user's answers in personality tests [17]. This allowed the authors to study how the accuracy of predictions increased with the number of dimensions of the Facebook likes. The method is linear in the PCA space and has the advantage that the results can be interpreted qualitatively. For example, young and female users could be predicted as liking "humorous and juvenile" (author's choice of words) statements such as, "I finally stop laughing . . . look back over at you and start all over again" [17].

The above method is potentially interesting in the context of stochastic parrots, because it allows us to, so to speak, look inside the parrot's brain. Large language models encode information using vector semantics: words and sentences are represented as vectors [14, 24, 20], referred to as embeddings. Words that occur in similar contexts tend to have similar meanings, therefore, they will have a similar vector [25]. The vectors are generally based on a co-occurrence matrix, a way of representing how often words co-occur. An alternative to using the term-document matrix to represent words as vectors of document counts, is to use the term-term matrix . If we then take every occurrence of each word and count the context words around it, we get a word-word co-occurrence matrix [14]. Embeddings can be obtained with transformers models [27, 9, 11, 13, 31, 30], which were initially developed for machine translation in 2017 [27, 28].

We can use principal components of the embeddings of a language model, with respect to a specific problem, in order to both understand what information is used in solving the task and to test the degree to which performance on that task is achieved from the representation of the data or from some other unknown mechanism. To make these statements concrete, we now outline what we do in this article. We address the task of predicting which character said which specific lines of dialogue in two US TV series: Big Bang Theory and Friends. This task is reminiscent of the personality research discussed above in that the characters in the show have very stereotypical personalities: can we predict character personalities from their line in the show? Such problems are of specific interest for this article, for three reasons (1) an increasing number of applications of AI involve supposed personality tests and analyses [10]; (2) such tests raise ethical issues about both reliability and applications [29, 1]; (3) they are sometimes used to imply that machines can understand us better than we understand ourselves [32]. The character personality test is an example of generalisation in the sense that, while large language models might have been fed data from these series, they haven't been trained to solve this specific task.

We proceed as follows. We first detail the method of and logistic regression on the principal components of the embeddings. We then analyse which PCA components are most predictive of statements by the characters, how the number components affects accuracy and differences between the TV shows. Finally, we compare performance of our simpler model to GPT-4 [22] and one human expert, with extensive experience of the two TV shows.

## 2 Methods

### 2.1 Embeddings and PCA

The dataset is the transcript of the first 10 seasons of the TV-series The Big Bang Theory [1] and 10 seasons of the TV-serie Friends [2] in English. We cleaned the dataset, by only keeping the main characters and their respective dialogue lines. This gives $44,966$ dialogue lines for the TV series The Big Bang Theory and $51,615$ dialogue lines for the TV series Friends. We then transformed these dialogue lines into a vector, i.e. we create embeddings using the python library SentenceTransformer and the model 'all-MiniLM-L6-v2' [26]. Each dialogue line then has a specific embedding, a vector of dimension 384. For comparison, the small text embedding of OpenAi, 'text-embedding-3-small', gives 1,536 output dimension [21, 5].

We then performed a principal component analysis (PCA) on the embeddings (for more details of the method we follow see [12]). Principal Component Analysis(PCA) determines the directions that maximize the variation in the data. The PCA is a procedure that takes dataset with several variables, to a smaller dataset with new variables (the principal components) that will be a linear combination of the former variables. Each dimension in this space corresponds to a feature that will be explicitly defined later. To ensure a representative view of the dataset, we need to standardize it so that no single variable disproportionately influences the analysis, by removing the mean then divide by the standard deviation. Then, we calculate the covariance matrix. A covariance matrix is a square matrix that shows the covariance between pairs of variables in the dataset. The diagonal of the matrix gives the variance of the variables and the other terms give the covariance between the pair of variables. The covariance measures of how much two random variables vary together, by estimating the linearity between them. From the covariance matrix we deduce the eigenvectors and eigenvalues, by doing an eigenvalue decomposition of the covariance matrix $\mathcal{C}$. We find the eigenvector by solving $(\mathcal{C} - \lambda Id)x = 0$, where $x$ is the eigenvector associated with the eigenvalue $\lambda$ The eigenvalue gives the magnitude (or accounted variance) of the data along the new feature dimension. The eigenvector gives the direction of the data along the new feature dimension, and forms the linear combination for a principal component. The eigenvalues are in descending order and as explained in [12], they 'maximize the explained variances on each dimension'. We refer to the the coefficients of the leading eigenvector as the first principal component (PCA1), the second eigenvector as PCA2 and so on. We reduce the 384 dimension of each embeddings to a dimension space of 300. All calculations were performed in Sklearn [23] and full code is available here [3].

An important aspect of our approach is gaining a qualitative understanding of how the principal components reflect the meaning of the dialogue lines. Each PCA corresponds to one eigenvector and consequently to one dimension from which we are able investigate which kind of phrases tied to that dimension. To help us make this analysis we used two-dimensional visualisation of the data. First we implemented a 10-means cluster on two principal components at a time, starting with the leading components (i.e PCA1 and PCA2). We colour each cluster and then assign the phrase nearest of the center as the cluster name (see figure 1). We also looked at the most extreme dialogue line in each PCA, by printing out the sentences with the highest values and the smallest values. From these we assigned a qualitative interpretation of the "meaning" of the leading PCAs. In the annex, we report the tenth highest values and the tenth smallest values.

### 2.2 Character prediction

In order to predict which dialogue line comes from which character, we use a logistic regression on the PCAs of the dialogue lines of the characters. We follow the notation from [12] and let $u_{j,i}$ be the

---

[1]http://www.kaggle.com/datasets/mitramir5/the-big-bang-theory-series-transcript
[2]https://github.com/yaylinda/friends-dialog/blob/master/data.csv
[3]https://github.com/amandinecaut/Friends_analysis.git

j-th the coefficient of the principal component of the i-th dialogue line. First, we normalise all the coefficients $u_{j,i}$ of the principal components by taking away the mean and dividing by the standard deviation, so each component has mean zero and standard deviation of one. We then performed a binomial logistic regression — e.g. does the dialogue line belong to Penny or Sheldon ? — based on a linear prediction of the dialogue line $i$:

$$\beta_0 + \beta_1 u_{1,i} + ... + \beta_n u_{n,i},$$

allowing to measure (using regression coefficients $\{\beta_0, ..., \beta_n\}$) how the the explanatory variables $u_{1,i}, ..., u_{n,i}$, impact the prediction. The fitted logistic regression is model is given by

$$\text{P (Sheldon|the i-th line is said by Sheldon or Penny)} = \frac{1}{1 + e^{-(\beta_0 + \beta_1 u_{1,i} + ... + \beta_n u_{n,i})}}$$

where $\beta_0$ determines the intercept (i.e. it is the outcome when all the other predictors variables are equal to zero). Each coefficient $\beta_i$ estimates the additional effect of adding the corresponding variable to the model prediction.

The sign of the coefficient indicates the influence of the specific principal component on the probability it is a particular character. If the sign is positive then it is more likely to be that character (Penny in the example above) if the dialogue line has larger and more positive values of that component. Conversely, if the sign is negative that means it is less likely to be that character if the dialogue line has larger and more positive values of that component. The larger the magnitude of the coefficient, the more important the predictor variable is in making the prediction.

For each TV series, we proceed to a logistic regression with 300 first PCAs, for each possible pair of characters. We obtain a predictor function and evaluate the absolute value of each regression coefficient. We obtain the magnitude of each coefficient and therefore assess which coefficients have the most importance in the logistic regression. Afterwards we take the ten regression coefficients with the largest aboslute value and plot them (see figure 3 and 8). From this analyse, we deduce which dimensions that have an impact on the character's prediction. To evaluate performance we calculate the AUC (Area Under The Curve) ROC (Receiver Operating Characteristics) curve to evaluate as a function of the dimensions.

## 2.3 Comparing to GPT4 and human expert

In order to test our method against a large language model we queried GPT4 with the following system prompt: *"You are expert on the TV series The Big Bang Theory. You are now being challenged to identify characters from the series. Try your best to do well. If you can beat another human expert there is a prize."* and a query that asked *"Tell me who was most likely out of Leonard and Sheldon (from the series Big Bang Theory) to have said the following line of dialogue: [DIALOGUE LINE]. Now state the most likely character as a single word, either Leonard and Sheldon. Do not write anything else."* Character and TV series names were adjusted appropriately for each test. We tested four pairs (Penny/Sheldon, Leonard/Sheldon, Phoebe/Ross, Phoebe/Chandler). We first repeated the above procedure 100 times, 50 times for each character, to test the accuracy of the classification (i.e. proportion of correct answers).

We also provided the same dialogue lines to two motivated human experts (who had watched both series in their entirety two times, most recently within the last year) and expressed a determination to beat GPT4. Both participants were relatives of the co-authors of this article. The same dialogue lines on which GPT4 was tested, were presented in a random order in the spreadsheet file. The subjects were asked to guess the name of the character for each dialogue line, and write it into the spreadsheet.

## 3 Results

### 3.1 Qualitative analysis of the principal components

We started by plotting the embedded dialogue lines 'Big Bang Theory' in terms of the six most important principal components, in order to visualise the most distinguishing features of the dialogue. The first two of these (PCA1 and PCA2) are shown in figure 1aa. The nearest neighbour clustering then allows us to see where different dialogue lines are found in these dimensions. We can see that larger negative values of PCA1 corresponds to very short phrases (for example 'Uh' in the pink

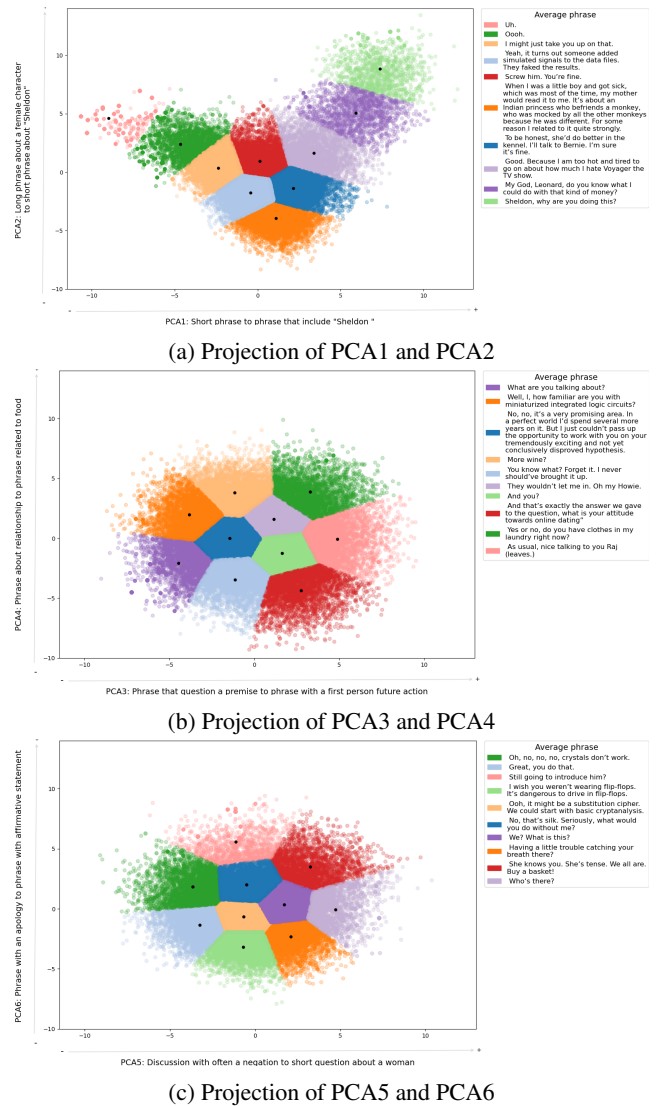

(a) Projection of PCA1 and PCA2

(b) Projection of PCA3 and PCA4

(c) Projection of PCA5 and PCA6

Figure 1: Projection of the first 6 PCAs. Each PCA has an interpretation from the qualitative analysis. Each plot has their respective cluster along with the average phrase of each cluster for The Big Bang Theory dialogue lines

cluster in the top left of the figure) and larger positive values of PCA1 correspond to phrases about Sheldon (for example 'Sheldon, what do you expect us to do?' in the green cluster in the top right of the figure). The qualitative analysis of PCA1 confirmed this pattern, with 'Yeah' being the most extreme negative value and 'You know, I was thinking. Without Sheldon, most of us would have never met, but Penny would still live across from him.' being the extreme positive value (see Annex 6 for a list of the ten most extreme positive and negative values of PCA1 and the other principal components).

Following the same approach for PCA2, we found that the negative values are associated with long phrases about a female characters an positive values with phrases about Sheldon. The most extreme negative value is 'Well, there was the time I had my tonsils out, and I shared a room with a little Vietnamese girl. She didn't make it through the night, but up till then, it was kind of fun.' and the most extreme positive value is 'Leonard, Sheldon.'(see annex 6). The cluster values in figure 1a also show the same pattern: with 'Indian princess who befriends a monkey who was mocked by all other monkeys because he was different. For some reason I related to it quite strongly' in the orange cluster

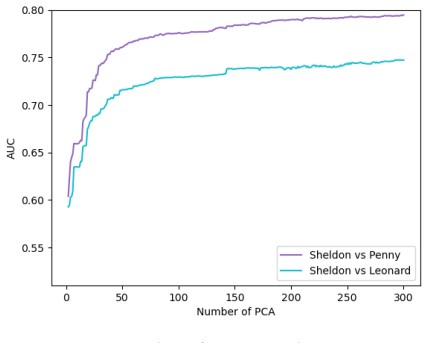
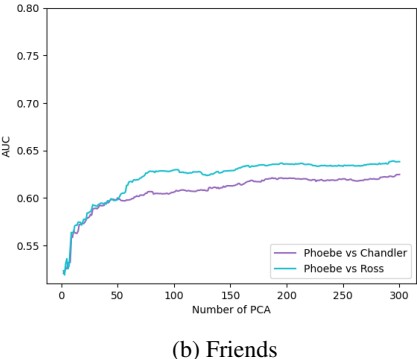

(a) The Big Bang Theory                        (b) Friends

Figure 2: AUC curves to assess the performance of the logistic regression, by increasing the number of dimensions, in the dialogue lines's prediction for two different couples for the two Tv serie

at the bottom of figure 1a and 'Sheldon, why are you doing this?' in the light green cluster at the top of the same figure.

A similar approach can be used to interpret figure 1b and c. PCA 3 ranges from phrase that questions a premise ('Really? I didn't know that.') to phrases with a first person future action ('Aw, sweetie, I'm comfortable around you, too.'). PCA 4 ranges from a phrase about relationship ('Really? That seems rather short sighted, coming from someone who is generally considered altogether unlikable. Why don't you take some time to reconsider?') to a phrase related to eating out ('Excellent! What are you planning to wear?'). The fifth dimension is phrase with often a negation or counterargument (like 'Oh no, no, no, crystals don't work', which is green in figure 1) to a short question about a woman (like 'She knows you. She's tense. We all are. Buy a basket!', which is red in the same figure). Finally, PCA 6 ranges from an apology (e.g. 'I wish you weren't wearing flip-flops. It's dangerous to drive in flip-flops') to a phrase with affirmative statement( e.g. 'Still going to introduce him?'). This final interpretation is even clearer when we look at the extreme negative value ( 'Relax, it wasn't your fault.') and extreme positive value ('Sure. I'd like to meet her.'). Overall, in The Big Bang Theory the distinguishing characteristics of the principal components often relate to the characters views of women. For Friends, there are also clear semantic differences in the sentences, although these appear to be less gender stereotyped. We give a full analysis of the leading six components in annex 5.3.

When we plot the average position of the characters in the space of the first two components, the differences are very small in comparison to the variation (figure 5 in annex 6). For example, while there is a distance of 0.33 between Leonard and Amy on the PCA 1 axis, the standard deviation of the values for the Leonard and Amy on that axis are 3.62 and 3.58, respectively. This observations indicates that it is impossible to distinguish the characters in terms of just a single dimension. We do note, though, that Friends characters are even closer together than The Big Bang Theory characters (the PCA1 distance between Chandler and Rachel is 0.15 and between Chandler and Joey is 0.11, while the standard deviations of Chandler, Rachel and Joey are respectively 3.62, 4.03 and 3.78). The biggest difference we observed is between Penny and Sheldon.

## 3.2 Character prediction

While a small number of principal component dimensions is not sufficient to tell the characters apart, can we use more of the dimensions to make the distinction? To test this we performed binomial logistic regression on pairs of characters as a function of the number of principal components we included in the model. The AUC values in figure 2a show a steady improvement in the predictions up to around 50 principal components for Big Bang Theory, after which only slight increases in performance are obtained. Sheldon and Penny were easier to distinguish using this method than Sheldon and Leonard. Figure 2b, shows that Friends characters were much more difficult to distinguish using this method.

If we view the principal component analysis as an attempt to capture the character's personality by their dialogue lines (as in the analysis by [32]) then we can say that the TV characters personality have a dimension of somewhere between 50 and 100. Each new dimension gives a small extra insight

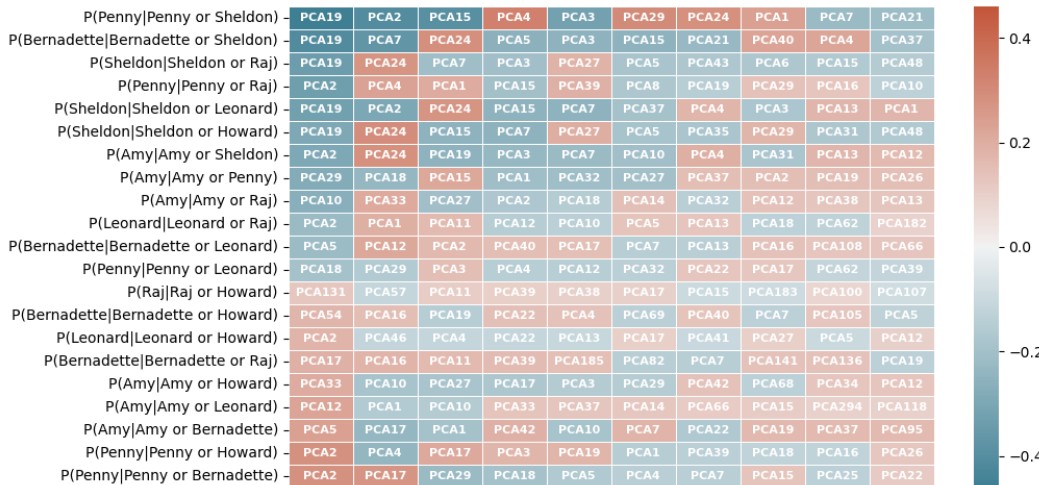

Figure 3: Regression coefficients for each possible character pairs for the TV series The Big Bang Theory. For each pair, we conduct a logistic regression to predict if the dialogue line is more likely to be said by a character1 such that $P$ (character1 $= 1|$the line is said by character1 or character2). We use the first 300 principal components in the logistic regression. Then, we assess the absolute value of each coefficient to determine their magnitude. Following this, we select the top ten coefficients for each linear predictor function. We report in this figure those coefficients, along with their corresponding dimensions. The coefficients are in decreasing order from left to right: the left side have the coefficient with the highest magnitude, the right side have the coefficients with the lowest magnitude.

into the character differences. Since Friends characters are more difficult to predict from what they say, we can conclude that Friends characters are less stereotyped than characters in The Big Bang Theory.

We can investigate which PCA dimensions best distinguish characters by looking at the coefficients of the binary regression. Figure 3 shows the ten most important components (determined by the magnitude of the absolute value of the coefficients in the regression) for distinguishing the characters dialogue lines in The Big Bang Theory. Each row represents a character pair, with the PCAs ordered from left to right according to the magnitude of the coefficients. The first column corresponds to the coefficient with the largest magnitude in the linear predictor function, the second column corresponds to the second coefficient with the second largest magnitude, and so on.

As an example, the first row is the character prediction for the couple 'Penny and Sheldon' should be read as considering the probability the dialogue line is by Penny, i.e. $P$ (Penny $= 1|$the line is said by Penny or Sheldon). The first cell entry, PCA19, is the coefficient in the linear predictor function with the largest absolute value. Performing a qualitative analysis on PCA19 (see annex 6) we find that negative coefficients correspond to lines about food and positive coefficients correspond to lines about comics. In this case, the coefficient of the PCA19 is negative, implying that if a dialogue line is about meal or food, it is more likely to be spoken by Penny than Sheldon.

The most common occurring component in figure 3 is exactly this PCA 19 (food vs. comics) which has 12 occurrences. PCA 2, which is long phrases about a female character versus phrases with one name has 11 occurrences. PCA 7 has 11 occurrences and ranges from phrases with yes/no to question about the current situation. The next most common occurring components are PCA4 (10 occurrences) which ranges from an apology to phrase with affirmative statement; PCA15 (10 occurrances) ranging from long phrases about a woman to short phrases about houses; PCA17 (9 occurrences) range from short phrases about travel to long food related phrases: PCA5 (9 occurrences) which range from long dialogue lines that express an opinion to short questions about a female character.

Figure 4 shows the relationship between the characters in terms of PCA19 (which distinguishes dialogue lines about meal/food related from those about comics). The graph shows the magnitude of

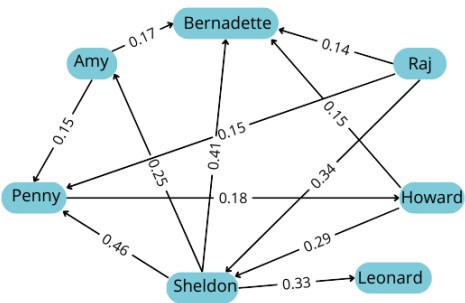

Figure 4: Relationship between characters in The Big Bang Theory in terms of PCA19 (that distinguishes lines about meal/food related form lines about comics). The value on the arrows show the magnitude of the coefficient. Only pairings where the absolute value of the regression coefficient is greater than 0.1 are included. The person at the start of the arrow talks about comics more than they talk about food compared to the person at the end of the arrow.

the coefficient and the direction of the arrow indicates that the coefficient is positive. For example, for P(Penny|the line is said by Penny or Sheldon) the regression coefficient for the PCA19 is negative, reflecting the fact that Penny talks more about food and Sheldon that talks more about comics, so the arrow points from Sheldon to Penny. Similarly, we see that Bernadette talks more about food than Raj, Howard Sheldon and Amy and thus the arrows point toward her. And Raj talks more about comics than Penny, Bernadette and even Sheldon, so the arrows point out from him in the figure. In the case of the TV series Friends, the magnitude of the regression coefficients are smaller than those for The Big Bang Theory and a more varied number of components are represented (see annex 5.3).

While the method for constructing figure 4 can give an indication of how the components distinguish the characters, we should bear in mind that in a regression of hundreds of variables (on which this graph is based) the relationships established are not always straightforward. For example, in the figure, we see that the respective models predict that Howard talks more about comics than Sheldon, who talks more about comics than Penny, and Penny talks more about comics than Howard. This inconsistency is likely due to other principal components distinguishing Penny and Howard better than PCA19, and PCA19 acting as a counterbalance, to these additional components. A full analysis of these relationships is beyond the scope of the current article.

## 3.3 Comparing to GPT4 and human expert

Initial prompting of GPT4 revealed that it has knowledge of the two TV series in its training data. GPT4 replied that it "can provide information about the show, its characters, plot points, cultural impact, and more". It was also able to provide motivation for its answers. For example, when we asked if this dialogue line 'Okay, sweetie, I don't know if we're gonna have cookies, or he's just gonna say hi, or really what's gonna happen, so just let me talk, and we'll...', it correctly answered 'Penny'. Then, when asked, it to explain why it draws conclusions about the characters, it cited criteria "Context of Character Behavior", "Speech Patterns" and "Interaction Dynamics".

For the set of 100 dialogue lines, a direct prompt to GPT4 (see methods for details) was correct for Penny versus Sheldon on 81 occasions, for Sheldon versus Leonard on 71 occasions, for Phoebe versus Ross on 66 occasions, and for Phoebe versus Chandler on 65 occasions. For these same test examples, the first human expert was correct on 71, 76, 67, and 59 occasions, respectively. The second human expert was correct on 74, 72, 70, and 73 occasions. For comparison, the accuracy (percentage correct over all sentences) for the 300 dimensional PCA model was 72.8%, 68.1%, 59.7% and 60.6% respectively. The standard error for a proportion of 70% is $0.7 \cdot 0.3 \cdot 100 \approx 4.5\%$, suggesting a comparable level of performance between the human experts and GPT4, and a slightly lower level of performance for the 300 dimensional PCA model.

## 4 Conclusion

Our qualitative analysis highlights how, when interpreted by a human, the principal components of the embeddings reflect the meaning of the dialogue lines of TV series. Many of dimensions contributing to the prediction are related to female characters. This can be attributed to the fact that the TV series portrays very stereotypical characters, with the main protagonists portrayed as geeks, embodying various clichés associated with them. A number of previous studies have identified gender and racial stereotyping within the way models represent data [4, 2, 29], we have shown that these dimensions are also important in the predictions these models make. Friends, in which the characters might be considered to have smaller stereotyped (within-group) differences, was more difficult to predict using this method.

We have shown that given the principal components of the dialogue in a TV series, we are able to predict the characters personality using logistic regression, to a level of performance slightly below that of GPT4. We needed 50-100 dimensions in the logistic regression to predict a dialogue line in the TV series. This might be said to support the idea of a language model more like a stochastic parrot than a spark of AI, in the sense that a large part of the predictive skill of the model can be obtained by adding up the components of the word embeddings and providing an appropriate prediction. Indeed, we have used a much smaller embedding vector (384 dimensions) that GPT4 (several thousand dimensions) to achieve somewhat comparable results.

That said, there remain two things which GPT4 does which our model does not. Firstly, our analysis starts from the sentence embeddings. Taking these embeddings as given ignores the complex process by which these are generated through training in the first place [9, 30]. Secondly, we had to specify the problem we wanted to solve as a logistic regression problem and train on previous data. GPT4, on the other hand, requires no additional training step and, from the given prompt, can identify the requested character. In light of these limitations, we see our work as highlighting the need to be more specific about claims related to sparks of AI [22]. We have shown that prediction part of the question of identifying TV character personality is (to some degree) obtainable from linear models, the question then is where the supposed spark lies? Is it in the creation of embeddings or is it in GPT4's ability to identify the prediction problem from the input provided by the user? We would suggest that further dissections of how these methods work, like we have done here for the prediction stage, can shed more light on these questions.

Our study is limited to a qualitative study of two very specific datasets. The contribution is primarily methodological. We propose an alternative to benchmark testing for understanding why a machine learning method works in the way it does, by comparing it to a method based on linear predictions. As such, it is a qualitative contribution to a larger debate around how to evaluate LLMs, rather than a quantitative demonstration of model performance.

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

# 5 Annex1 : Supplementary material

## 5.1 Average Position for each main character of the two Tv series

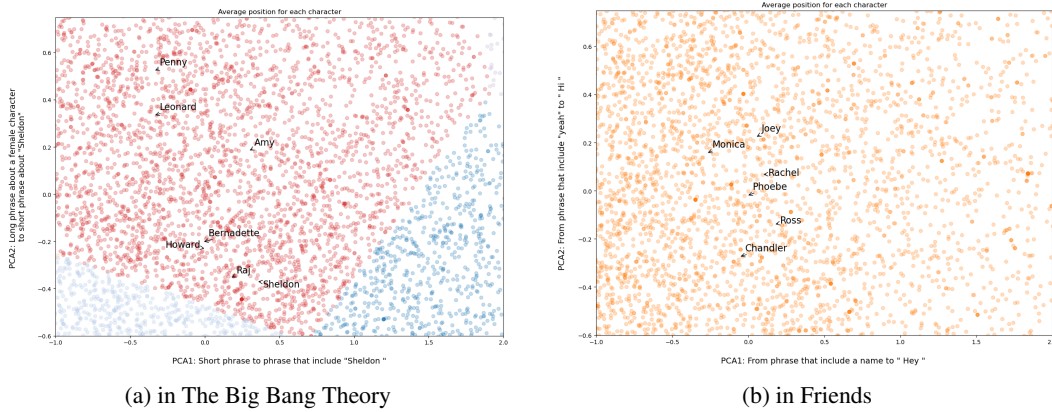

(a) in The Big Bang Theory        (b) in Friends

Figure 5: Projection of the two first PCAs, and their respective interpretation, with the average position for each main character of the two Tv series

## 5.2 Accuracy curves the two Tv serie

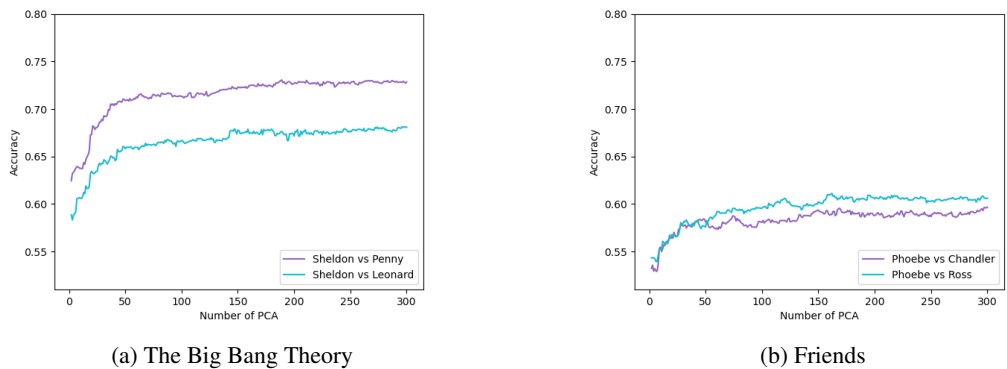

(a) The Big Bang Theory        (b) Friends

Figure 6: Accuracy curves to assess the performance of the logistic regression, by increasing the number of dimensions, in the dialogue lines's prediction for two different couples for the two Tv serie

## 5.3 Friends Analysis

For Friends, we analyse similarly the 6 first dimensions as seen in Figure 7. The PCA1 is interpreted as phrase that include a name to 'Hey'. This is illustrate with the figure 7a by the dark blue cluster in the left with the average phrase 'Ms. Monroe... Oh there you go', for the negative larger values of the PCA1, and by the red cluster on the top right with average phrase 'Hey' for the positive larger values of the PCA1. The qualitative analysis, in annex 7, gives as the most extreme negative value of the PCA1 the phrase 'Yeah. It's just gonna be too hard. Y'know? I mean, it's Ross. How can I watch him get married? Y'know it's just, it's for the best, y'know it is, it's... Y'know, plus, somebody's got to stay here with Phoebe! Y'know she's gonna be pretty big by then, and she needs someone to help her tie her shoes; drive her to the hospital in case she goes into labour.'. The most extreme positive value of the PCA1 is 'Hey'. The qualitative confirm our earlier statement about the interpretation of the PCA1.

The PCA2 is phrase that include 'yeah' to 'Hi'. The negative values of the PCA2 can be found on the figure 7a, for example from the light green cluster of at the bottom left, with average phrase 'Um,

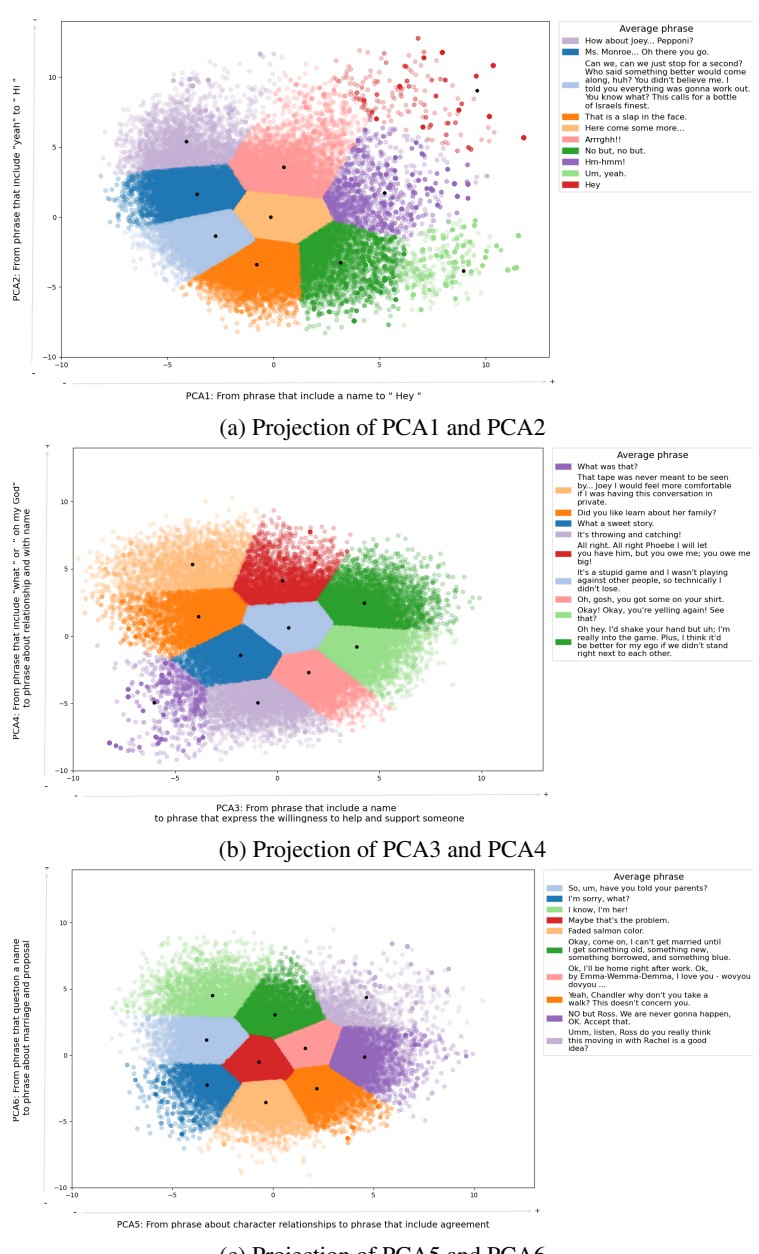

(a) Projection of PCA1 and PCA2

(b) Projection of PCA3 and PCA4

(c) Projection of PCA5 and PCA6

Figure 7: Projection of the first 6 PCAs. Each PCA has an interpretation from the qualitative analysis. Each plot has their respective cluster along with the average phrase of each cluster for Friends dialogue lines

yeah.', and the positive value are on the top red cluster with average phrase 'Hey'. It is confirm from the qualitative analysis in annex 7, give the most extreme negative value 'Yeah, fair enough.' and the most extreme positive value 'Hey! Hi!'.

The PCA3 is phrase that express the willingness to help and support someone to phrase that include a name.The projected values of the PCA3 are on the figure 7b, the negative values are on the left of the graph, for example, the orange cluster with average phrase 'Did you like learn about her family?'. In regards of the positive values, they are on the right of the graph, for example the light green cluster with average phrase 'Okay! Okay, you're yelling again! See that?'. The qualitative analysis, see annex 7, shows the most extreme negative value of the PCA3 is 'Phoebe?! Wait a-but-but she just, she said that Joey was her backup.' and the most extreme positive value is ' Hi! I'm back. Yeah, that sounds great. Okay. Well, we'll do it then. Okay, bye-bye.'

The PCA4 interpretation is about phrase that include 'what' or 'oh my God', for example in the figure 7b in the dark purple cluster in the bottom left with average phrase 'What was that?', to phrase about relationship and with the name, for example in the figure 7b with the dark green cluster with average phrase 'Oh hey, I'd shake your hand but uh: I'm really into the game. Plus, I think it'd be better for my ego if we didn't stand rigt to each other.'. The qualitative analysis, in annex 7, confirm our statement with the following most extreme negative value 'What?! What is it?!. and the most extreme positive value 'Well it's okay. Chandler is talking to her.'

PCA5 is phrase about character relationship to phrase that include agreement. As seen in the figure 7c, the negative value of PCA5 are represented on the graph on the left, for example with the light blue cluster with average phrase 'So, um, have you told your parents?'. The positive value of PCA5 are on the right of the figure 7c, as we can pick out from the dark purple cluster with average phrase 'No, but Ross. We are never gonna happen, OK. Accept that.'. The qualitative analysis verify our interpretation, in annex 7, we see that the most extreme negative value is the phrase 'But, also, what happened between you and your Mom?. and the most extreme positive value is 'Yeah! That would be great!'.

We interpret the PCA6 as phrase that question a name to phrase about marriage and proposal. The PCA6 projection is illustrate in the figure 7c, with negative values as the bottom, with for example the cluster dark orange with average phrase 'Yeah, Chandler why don't you take a walk? This doesn't concern you.'. The positive value of the PCA6 are in the top of the graph, for example dark green cluster with average phrase 'Okay, come on, I can't get married until I get something old, something new, something borrowed, and something blue'. Our statement confirmed by the qualitative analysis, in annex 7, with the most extreme negative value is the phrase 'Wait a minute. What's his name?' and the most extreme positive value is the phrase 'Yes! We're getting married?!'.

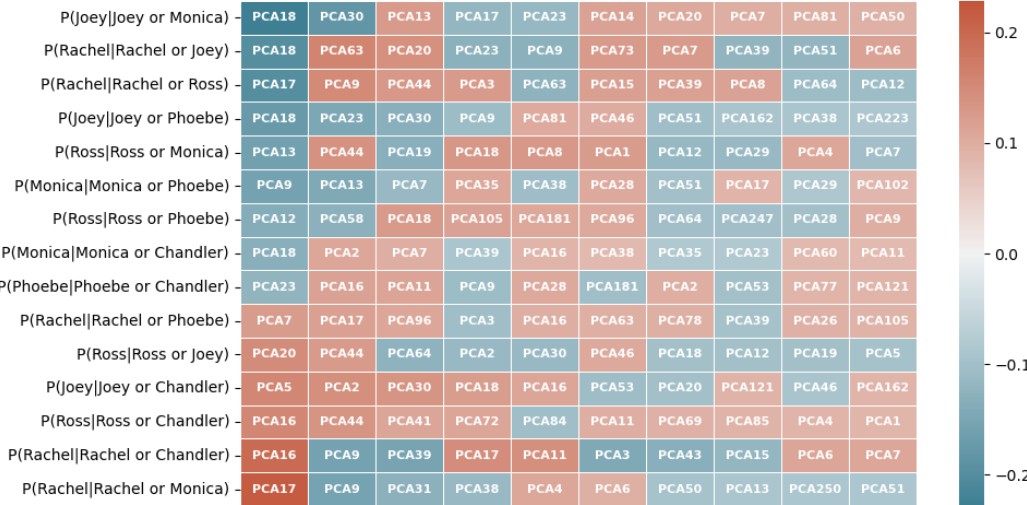

Figure 8: Regression coefficients for each possible character pairs for the TV series Friends. For each pair, we conduct a logistic regression to predict if the dialogue line is more likely to be said by a character1 such that $P$ (character1 = 1|the line is said by character1 or character2). We use the first 300 principal components in the logistic regression. Then, we assess the absolute value of each coefficient to determine their magnitude. Following this, we select the top ten coefficients for each linear predictor function. We report in this figure those coefficients, along with their corresponding dimensions. The coefficients are in decreasing order from left to right: the left side have the coefficient with the highest magnitude, the right side have the coefficients with the lowest magnitude. The rows are arrange such that the first row (the most significant coefficients) is in increasing order

In the case of the TV series Friends, in the figure 8, the first column are the most significant regression coefficient for each pair. We can notice that the most extreme negative value is in the first row and belongs to the regression coefficient of the character's dialogue lines prediction between Joey and Monica. The probability is as follow, $P$ (Joey = 1|the line is said by Joey or Monica) = $p$ and $P$ (Monica = 0|the line is said by Joey or Monica)) = $1 - p$. The corresponding dimension of the first coefficient is the PCA 18, it depicts phrase from 'Oh no' to phrase that include 'yeah' or 'look' (see qualitative analysis in annex **??**. In other words, a phrase that include 'Oh no' is more likely from Joey. The most extreme positive value in this first column appears in the last row, corresponding to the regression coefficients for predicting dialogue lines between the pair 'Rachel and Monica'. The probability is such that $P$ (Rachel = 1|the line is said by Rachel or Monica) = $p$ and $P$ (Monica = 0|the line is said by Rachel or Monica) = $1 - p$. The coefficient correspond to the dimension PCA17: from phrase that include 'Joey' to phrase that include 'Ross'. We can deduce that, if a phrase include 'Ross' it is more likely from Rachel.

| PCA 9 | 8 occurrences | From phrase that include 'Oh', to question about what the people has been doing |
| PCA 18 | 8 occurrences | From 'Oh no' to phrase that include 'yeah' or 'look' |
| PCA 7 | 7 occurrences | From phrase which is an answer a statement to 'What?' |
| PCA 17 | 6 occurrences | From phrase that include 'Joey' to phrase that include 'Ross' |
| PCA 16 | 6 occurrences | From phrase about a statement on a character to question with 'What' |

Table 1: Interpretation of the most important dimension in the dialogue lines prediction in Friends, with the number of time they occurs in the figure 8

For Friends, we also count the occurrences of each PCA from the figure 8, and then interpret them. We recapitulate the information in the table 1. Contrary to The Big Bang Theory the phrases in Friends are much shorter, more exclamatory, and there are less obvious topic like food or comics.

In the TV series Friends, we note fewer instances of the main principal component analysis. For instance, in The Big Bang Theory, PCA19 occurs most frequently, appearing 12 times. However, in Friends, PCA9 and PCA18 are the most common dimensions, each occurring 8 times. If we count the number of different PCA in figure 3 for The Big Bang Theory we obtain 59, and 56 different PCA for Friends in the figure 8. The number of dimension is similar in both case, but we can pick out that the magnitude of the coefficient is slightly higher in The Big Bang Theory than in Friends. Since the TV serie Friends has less occurrences of the main PCAs, smaller magnitude in the regression coefficients and less AUC accuracy, therefore more dimension are needed into the dialogue line predictions. This is visible on the figure 5, where we can see that average position of the character in Friends are more closer than the average position of the character in The Big Bang Theory.

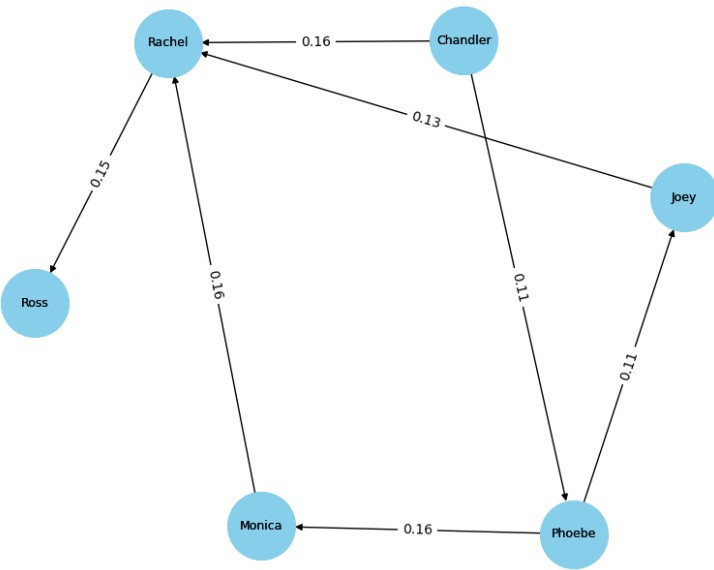

Figure 9: Relationship between characters in Friends for the dimension that occurs the most in Figure 9 (PCA9) with phrase that include 'Oh', to question about what the people has been doing. The person at the start of the arrow ask more about what the people has been doing more than they have phrase that include 'Oh' to the person at the end of the arrow.

In the figure 9, we show the relationship between the character of Friends for the PCA9, the dimension that have the most occurrences, it is interpret as with phrase that include 'Oh' to question about what the people has been doing. For example, in the dialogue lines prediction $P(\text{Rachel} = 1|\text{the line is said by Rachel or Ross})$, the regression coefficient is positive, then if it is a question about what the people has been doing, it is more likely from Rachel, and if it is a phrase that include 'Oh', then it is more likely to be from Ross. Then the arrow goes from Rachel to Ross. If the regression coefficient is negative, for example when we want to predict a dialogue line such that $P(\text{Rachel} = 1|\text{the line is said by Rachel or Joey})$, then if a phrase include 'Oh' it is more likely to be said by Rachel, and if it is a question about what the people has been doing, it is more likely from Joey. The arrows goes from Joey to Rachel.

# 6 Annex 2: Dialogue example of The Big Bang Theory

## 6.1 PCA1

### 6.1.1 Lowest coefficient

SHELDON : Yeah.

LEONARD : Yeah.

LEONARD : Yeah.

LEONARD : Yeah.

SHELDON : Yeah.

LEONARD : Yeah.

PENNY : Yeah.

PENNY : Yeah.

PENNY : Yeah.

SHELDON : Yeah.

### 6.1.2 Highest coefficient

BERNADETTE : You know, I was thinking. Without Sheldon, most of us would have never met, but Penny would still live across from him.

AMY : Which couldn't have happened if you didn't live across the hall from her, which couldn't have happened without Sheldon. Same goes with you guys. If Leonard wasn't with Penny, she never would have set you up.

PENNY : Oh, my God, Sheldon the genius is jealous of Leonard.

HOWARD : Now, I never thought I'd say this, but I'm kind of excited to see Sheldon.

AMY : This isn't about me and Sheldon. This is about Rajesh moving in with Leonard and Penny.

RAJ : It's a human emotion, Sheldon. Everyone gets jealous. I'm jealous of Leonard and Penny and Howard and Bernadette for being in such happy relationships.

LEONARD : Oh, come on. Sheldon, have you ever once heard me say that I don't trust Penny? Sheldon? Where did he go?

PENNY : Well, yeah, he'd been living with Sheldon.

LEONARD : Really. Who do you think did that, Sheldon?

AMY : Well, I was hoping the next person I dated would be a little less like Sheldon.

## 6.2 PCA2

### 6.2.1 Lowest coefficient

AMY : Well, there was the time I had my tonsils out, and I shared a room with a little Vietnamese girl. She didn't make it through the night, but up till then, it was kind of fun.

BERNADETTE : Because it would make you seem like something she already thinks you are.

BERNADETTE : You don't think she'd actually send you something gross or dangerous, do you?

LEONARD : Too expensive. You'd think I'd be used to women withholding their love. I
mean, my mother did. I mean, no matter how hard I tried, she just didn't have any
interest in me.

LEONARD : I mean, I know she's not my girlfriend or anything, but wouldn't you think
she'd feel a little bad that I'm going to be gone for the whole summer?

SHELDON : Or you might think she thinks you think it's a date even though she doesn't.

LEONARD : Yeah, yeah, that's the fun part. We're also getting new curtains for my
bedroom, and a dust ruffle, and a duvet, and I don't even know what a duvet is
but I'm pretty sure if I did I wouldn't want one, but every time I talk to her about
moving out she cries and we have sex.

AMY : Parental pressure can be daunting. I remember the battle with my mother about
shaving my legs. Last year, I finally gave in and let her do it.

LEONARD : Don't you think if a woman was living with me I'd be the first one to know
about it?

SHELDON : I was hoping she might listen to you about the dangers of owning unhygienic
furniture.

### 6.2.2 Highest coefficient

LEONARD : Leonard, Sheldon.

LEONARD : Hi, I'm Leonard, this is Sheldon.

HOWARD : What about Sheldon?

LEONARD : Sheldon. . .

LEONARD : Sheldon. . .

LEONARD : Sheldon. . .

LEONARD : Sheldon. . .

HOWARD : Sheldon.

LEONARD : Sheldon.

HOWARD : Sheldon.

## 6.3 PCA3

### 6.3.1 Lowest coefficient

SHELDON : Really? I didn't know that.

PENNY : Did they make a movie about it?

RAJ : How did that even happen? Did they know that's what they were doing when they
were doing it?

HOWARD : Yeah, I saw it on Mythbusters.

BERNADETTE : Do they have that?

SHELDON : A more plausible explanation is that his work in robotics has made an
amazing leap forward.

SHELDON : Oh. Is it true they used scuba gear to create the sound of Darth Vader
breathing?

HOWARD : Not exactly. They spent a ton of money developing this dandruff medication
that had the side effect of horrible anal leakage.

SHELDON : It's been around for 25 years, and has been extensively corroborated by
other researchers.

584    RAJ : Did he get superpowers?

### 6.3.2 Highest coefficient

586    PENNY : Aw, sweetie, I'm comfortable around you, too.

587    LEONARD : Great. Just relax and enjoy. Tonight is all about you.

588    SHELDON : Thank you, but I'll be fine.

589    PENNY : Okay, well, we'll talk to you guys later. Bye. She said not to come. It's gonna
590        be a while.

591    SHELDON : Fine, let's go. Thank you for letting me sleep on your couch.

592    SHELDON : Oh, well, you two sit down and get to know each other. I'll get your room
593        ready.

594    AMY : I will. I wish you were here.

595    LEONARD : Let's go. Okay, you two, just, have a nice... whatever this is.

596    PENNY : All right. Well, you guys have fun. I guess I'll see you Sunday night.

597    LEONARD : Yeah, no, I'm fine. It's good, it's a good party, thanks for having us, it's just
598        getting a little late so. . . .

## 6.4 PCA4

### 6.4.1 Lowest coefficient

601    SHELDON : Really? That seems rather short sighted, coming from someone who is
602        generally considered altogether unlikable. Why don't you take some time to
603        reconsider?

604    SHELDON : Yes, and she's not taking my feelings into account at all. Maybe it's time I
605        teach her a lesson.

606    AMY : No, we're sorry. We never should have been comparing relationships in the first
607        place.

608    HOWARD : Yeah, she was dating this guy, and I was kind of a jerk to her about it.

609    SHELDON : Yeah, but to be fair, he only said the part about him getting sick of you.

610    SHELDON : Oh, you're right. I could never be with a woman whose self-esteem was so
611        low she'd be with Leonard.

612    SHELDON : Not true. No, look at me. I had an engagement ring to give a girl, and
613        instead, she rejected me. And am I emotional about that? No. No, I am sitting
614        here on a couch, talking about my favourite TV character like nothing happened.
615        'Cause I am just like him, all logical, all the time.

616    SHELDON : It hurts that you would lie to me, Amy. I thought our relationship was based
617        on trust and a mutual admiration that skews in my favour.

618    PENNY : Okay, I have not tried to change Leonard. That's just what happens in relation-
619        ships. Look how much Amy's changed you.

620    PENNY : I get that, okay? It's just, Leonard and I have been married for two years, and
621        we're no further along than when we were dating.

### 6.4.2 Highest coefficient

623    SHELDON : Excellent! What are you planning to wear?

624    HOWARD : In our new minivan. Hey, what's for lunch?

625    BERNADETTE : Where are you guys going to eat?

PENNY : What beverage do you make for that?

SHELDON : Oh, I have quite the evening planned. Our foetus-friendly festival of fun begins with an in-depth look at the world of model trains, and then we'll kick things up a notch and explore all the different ways that you can make toast.

LEONARD : What are you drinking there? A little eggnog?

RAJ : Sounds great!

SHELDON : In here, you'll find emergency provisions. An eight-day supply of food and water, a crossbow, season two of Star Trek: The Original Series on a high-density flash drive.

AMY : I'm going to the vending machine. Do you want anything?

SHELDON : Greetings, gentlemen. How goes your little project?

## 6.5  PCA5

### 6.5.1  Lowest coefficient

BERNADETTE : Absolutely. All we need to do is spend a little time and find something you're passionate about.

PENNY : Okay, a simple yes will do.

BERNADETTE : Of course you can. But maybe a good rule would be to wait for people to bring it up.

RAJ : No, no, it's a very promising area. In a perfect world I'd spend several more years on it. But I just couldn't pass up the opportunity to work with you on your tremendously exciting and not yet conclusively disproved hypothesis.

LEONARD : Sheldon, I think this will work. Let's just try it my way.

LEONARD : If that's what you want to do, yes.

HOWARD : Yeah, this is a bad idea. We should go.

AMY : Of course. I get to be part of the first team to use radon markers to map the structures that...

PENNY : Yeah. And there are a few things we need to stay on top of. So we thought it would useful, and I can't believe I'm about to say this, um.

LEONARD : No, I don't want to do it. You can do it.

### 6.5.2  Highest coefficient

HOWARD : How was she?

LEONARD : When was the last time you saw her?

LEONARD : How's your mom holding up?

AMY : Oh. What was her name?

LEONARD : How's she doing?

BERNADETTE : It was your mom.

LEONARD : Aw. What's wrong with her?

HOWARD : My mom died.

SHELDON : What's her name?

HOWARD : So, what is she doing today?

## 6.6 PCA6

### 6.6.1 Lowest coefficient

LEONARD : Relax, it wasn't your fault.

HOWARD : I'm sorry, too. It's all my fault.

AMY : Well, I didn't, and it's your fault.

PENNY : I'm sorry I yelled at you. It's not your fault.

LEONARD : It's not your fault.

AMY : It's not your fault.

SHELDON : It's simple biology. There's nothing I can do about it.

HOWARD : Look, I have felt terrible about this for years, and I'm glad I have the opportunity to tell you just how sorry I am.

LEONARD : This time, it's your fault.

LEONARD : Well, that's not your fault.

### 6.6.2 Highest coefficient

LEONARD : Sure. I'd like to meet her.

LEONARD : Will Amy be joining us for dinner?

BERNADETTE : Maybe, if she asks.

HOWARD : Sure she would. Ma, do you mind if Bernadette stays here this weekend?

LEONARD : No, no, of course not. Just have your relationship someplace else.

SHELDON : I'm going to find her and ask her to marry me. And if she says yes, we can put this behind us and resume our relationship. And if she says no, well, then she can just ponfo miran.

HOWARD : Yes!

HOWARD : Yes!

HOWARD : Yes!

HOWARD : Yes!

## 6.7 PCA19

### 6.7.1 Highest coefficient

SHELDON : Yes. Oh, I'm so excited. And I just can't hide it.

PENNY : I do, it's just he wants to go to that party at the comic book store. A lot of the guys that hang out there are kind of creepy.

LEONARD : Oh, I'm just trying to find the stupid next of kin to this stupid video store owner so I can return the DVD and see the look on Sheldon's stupid face when he sees that I didn't let this get to me.

HOWARD : Ooh, I want to go to the comic book store. (He leaves.)

PENNY : Yeah, but those tickets only get him into Comic-Con. That dress gets me into anywhere I want.

PENNY : No, come on, it's going to be fun, and you all look great, I mean, look at you, Thor, and, oh, Peter Pan, that's so cute.

BERNADETTE : Is it me, or is there something fun about watching him just float there?

706 HOWARD : Come on, Sheldon, there's so few places I can wear my jester costume.

707 RAJ : So, listen to what he wrote. Uh, I saw you play at the comic book store. You guys
708 rock. And then there's an animated smiley face raising the roof like this.

709 SHELDON : Oh no! (He is also wearing a Flash costume.)

### 6.7.2 Lowest coefficient

711 SHELDON : We can't have Thai food, we had Indian for lunch.

712 SHELDON : It was a Monday afternoon. You joined us for Indian food.

713 SHELDON : Good morning, Friend Howard. Friend Raj. I see you gentlemen are
714 enjoying beverages. Perhaps they would taste better out of these.

715 RAJ : My stomach. Indian food doesn't agree with me. Ironic, isn't it?

716 LEONARD : Well the only way we can play teams at this point is if we cut Raj in half.

717 LEONARD : I've always been a little confused about this. Why don't Hindus eat beef?

718 RAJ : Of course, but it's all Indian food. You can't find a bagel in Mumbai to save your
719 life. Schmear me.

720 SHELDON : Yeah, I actually have information about Raj that would be helpful with this
721 discussion.

722 RAJ : We Indians invented them. You're welcome.

723 LEONARD : Here's an idea, why don't we just go out for Indian food.

## 6.8 PCA7

### 6.8.1 Highest coefficient

726 RAJ : He's gonna be here any second, what should we do?

727 PENNY : What are you guys gonna do?

728 LEONARD : What are we gonna do?

729 HOWARD : What are we gonna do?!

730 AMY : What's going on with him?

731 LEONARD : What are we going to do?

732 RAJ : So what are we going to do tonight?

733 LEONARD : What's with him?

734 HOWARD : What's with him?

735 PENNY : What's with him?

### 6.8.2 Lowest coefficient

737 PENNY : Oh, Sheldon, are these letters from your grandmother?

738 PENNY : I do, and you know, I don't think I've ever thanked you properly for helping
739 me get it.

740 SHELDON : Oh, yes. In fact, I improved upon it.

741 SHELDON : No, of course not. No, I used trickery and deceit.

742 LEONARD : Yeah, no, I do, I use those... uh... just to polish up my... spear-fishing
743 equipment. I spear fish. When I'm not crossbow hunting, I spear fish. Uh, Penny,
744 this is Sheldon's twin sister, Missy. Missy, this is our neighbour Penny.

745 LEONARD : Yes, I've always admired that about you.

746    PENNY : She was right, you know. The locus of my identity is totally exterior to me.

747    LEONARD : Oh, yes. Indeed, I did.

748    LEONARD : No, no, I'm good. If my P.E. teachers had told me this is what I was training
749    for, I would have tried a lot harder.

750    RAJ : Do you kind of look like a shiny Sheldon?

## 6.9    PCA15

### 6.9.1    Highest coefficient

753    BERNADETTE : Yeah. You're inviting him into your home. It's intimate. It's where your
754    underpants live.

755    RAJ : It's a lease.

756    LEONARD : What was I supposed to do? He needed a place to sleep it off.

757    LEONARD : Ask him for a napkin, I dare you. (There is a knock on the door.) I'll get it.

758    RAJ : He probably just goes to the bathroom.

759    HOWARD : Maybe the problem is he thinks you're available. Does he know you're
760    dating Sheldon?

761    LEONARD : What if he lives in your garage?

762    HOWARD : How'd you get him to come to your house?

763    BERNADETTE : What are you going to do? Doesn't he know you have a boyfriend?

764    LEONARD : He's in his bedroom.

### 6.9.2    Lowest coefficient

766    LEONARD : Look, do I think that you are talented and that you are beautiful? Of course I
767    do. But isn't Los Angeles full of actresses who are just as talented, just as beautiful?
768    All right, look, we'll come back to that.

769    AMY : I do. Penny, Bernadette and I are sorry.

770    RAJ : Oh, yes, we've got the moon and the trees and Elizabeth McNulty, who apparently
771    died when she was the same age I am.

772    SHELDON : And on a different, but not unrelated topic, based on your current efforts to
773    buoy my spirits, do you truly believe that you were ever fit to be a cheer leader?

774    SHELDON : Hello, female children. Allow me to inspire you with a story about a great
775    female scientist. Polish-born, French-educated Madame Curie. Co-discoverer of
776    radioactivity, she was a hero of science, until her hair fell out, her vomit and stool
777    became filled with blood, and she was poisoned to death by her own discovery.
778    With a little hard work, I see no reason why that can't happen to any of you. Are
779    we done? Can we go?

780    SHELDON : No, I don't think so. Those dolls represent three things I do not care for,
781    clowns, children and raggediness. I think it's a lost cause.

782    SHELDON : Yes. I think prolonged exposure to Penny has turned her into a bit of a
783    Gabby Gertie.

784    RAJ : Yes, isn't she an amazing actress.

785    SHELDON : Actually, I thought the first two renditions were far more compelling. Previ-
786    ously I felt sympathy for the Leonard character, now I just find him to be whiny
787    and annoying.

788    HOWARD : She was just so sad all the time. I was the only person who could cheer her
789    up. Well, me and Ben and Jerry.

## 6.10 PCA17

### 6.10.1 Highest coefficient

SHELDON : Penny, a moment. We just had Thai food. In that culture, the last morsel is called the krengjai piece, and it is reserved for the most important and valued member of the group.

LEONARD : Yeah, it's delicious, the sarcasm's a little stale, though. Hey, how about this? Until we figure out what to do with the ring, Penny holds on to it.

PENNY : Okay, sweetie, I don't know if we're gonna have cookies, or he's just gonna say hi, or really what's gonna happen, so just let me talk, and we'll...

PENNY : Fine. What do you want?

HOWARD : Okay, this one is for a Cadbury Creme Egg.

LEONARD : Ah, well, what's this? A pot of oatmeal? Or, thanks to you, what I will now call gloatmeal.

SHELDON : I'm sorry, but these are just ordinary foods with the names bent into tortured puns. The dishes themselves are in no way Halloweenie.

LEONARD : Ah, that's a good question. Apparently someone was being awfully flirty while not wearing their engagement ring, causing another someone to show up here thinking the first someone might be available.

PENNY : Okay, well, I'd offer you Halloween candy, but that's gone. So, what's up?

RAJ : Okay. Shall we? Oh, my God. It's light, it's flaky, it's buttery. You don't need to have sex with him, just eat one of these.

### 6.10.2 Lowest coefficient

RAJ : Then she's going to have to convince your mother to let you go into space.

HOWARD : Then get out of my house.

BERNADETTE : Yeah, if you want to go off the grid, you have to move out of your mother's house.

SHELDON : I can't believe my own mother is abandoning me.

HOWARD : I will. I'm obviously not going to live in my mother's house for the rest of my life. I'm not a child.

LEONARD : With your career?

BERNADETTE : You're a real hero, Howard.

BERNADETTE : I'm proud of her. This is a great opportunity. It's nice to see her take it seriously.

LEONARD : Also instead of just living in your mother's house, you could actually live inside her body.

LEONARD : And now you're also an astronaut.

## 7 Annex 3: Dialogue example of Friends

### 7.1 PCA1

#### 7.1.1 Highest coefficient

CHANDLER : Hey.

CHANDLER : Hey.

PHOEBE : Hey.

RACHEL : Hey.

ROSS : Hey.

MONICA : Hey.

RACHEL : Hey.

RACHEL : Hey.

CHANDLER : Hey.

ROSS : Hey.

#### 7.1.2 Lowest coefficient

RACHEL : Yeah. It's just gonna be too hard. Y'know? I mean, it's Ross. How can I watch him get married? Y'know it's just, it's for the best, y'know it is, it's... Y'know, plus, somebody's got to stay here with Phoebe! Y'know she's gonna be pretty big by then, and she needs someone to help her tie her shoes; drive her to the hospital in case she goes into labour.

RACHEL : Ross, you know what? She may need one..We're just going to have to make our peace with that! Monica and Chandler's apartment.

JOEY : Look we've got to find her. Phoebe just called!! Rachel's coming to tell Ross she loves him!!

CHANDLER : Well, she's just so much fun with Joey, I just assumed, she'd still be living with him.

JOEY : Well, remember when they got in that big fight and broke up and we were all stuck in her with no food or anything? Well, when Ross said Rachel at the wedding, I figured it was gonna happen again, so I hid this in here.

MONICA : I can't believe this. Rachel and Joey?

RACHEL : Look Monica, getting cold feet is very common. Y'know, it's-it's just because of all the anticipation and you just have to remember that you love Chandler. And also, I ran out on a wedding. You don't get to keep the gifts.

MONICA : No, look, she's obviously unstable, okay? I mean she's thinking about running out on her wedding day. Okay, fine! But I mean, look at the position she's putting him in! What's he gonna do? Ross is gonna run over there on the wedding day and break up the marriage?! I mean, who would do that?! Okay, fine, all right, but that's y'know, it's different! Although it did involve a lot of the same people.

PHOEBE : Why do you think, she's having so much fun living with Joey?

PHOEBE : It's so weird seeing Ross and Rachel with a baby. It's just so grown up.

### 7.2 PCA2

#### 7.2.1 Highest coefficient

RACHEL : Hey! Hi!

868  RACHEL : Hey! Hi!

869  ROSS : Hey! Hi!

870  PHOEBE : Hey! Hi!

871  RACHEL : Hey! We're here!

872  RACHEL : Hi!!

873  RACHEL : Hi!!

874  MONICA : Hi!

875  MONICA : Hi!

876  MONICA : Hi!

877  ### 7.2.2  Lowest coefficient

878  RACHEL : Yeah, fair enough.

879  RACHEL : Really? You think so?

880  PHOEBE : Really? You think?

881  PHOEBE : Yeah, what's your point?

882  PHOEBE : Yeah, but not just that.

883  RACHEL : No, you're right, you are absolutely right. I mean that makes, that makes
884      everything different.

885  JOEY : No. Really?

886  ROSS : Really? Its not just frowned upon?

887  JOEY : Yeah, I wouldn't know about that.

888  CHANDLER : Yeah, you're right about that.

889  ## 7.3  PCA3

890  ### 7.3.1  Highest coefficient

891  CHANDLER : Hi! I'm back. Yeah, that sounds great. Okay. Well, we'll do it then. Okay,
892      bye-bye.

893  ROSS : I'll do it. Hey, whatever you need me to do, I'm your man. Whoa-oh-whoa! Are
894      you, are you okay?

895  RACHEL : No, come on, I'm totally ok. I don't need you to come! I can totally handle
896      this on my own.

897  ROSS : I'll help you. Yeah, I'll make up a schedule and make sure you stick to it. And
898      plus, it'll give me something to do.

899  JOEY : Alright, alright. I'm around. Go ahead.

900  PHOEBE : Anyway, I should go. Okay, bye.

901  MONICA : Ok first of all...It would be great. But that's not what I'm here to talk to you
902      about. I need to borrow some money.

903  MONICA : No, I'll do it. You just stick to your job.

904  ROSS : Oh, that'd be great! Okay, but if you do, make sure it seems like you're there to
905      see him, okay, and you're not like doing it as a favour to me.

906  JOEY : Sure, yeah. I don't have time to say thank you because I really gotta go.

### 7.3.2 Lowest coefficient

RACHEL : Phoebe?! Wait a-but-but she just, she said that Joey was her backup.

MONICA : They thought Joey was a child?

CHANDLER : And then Joey remembered something.

RACHEL : I thought it was Chandler!

MONICA : Does it have to do with Joey?

RACHEL : Joey! Why did you tell Chandler that Monica was getting a boob job?

MONICA : And Rachel. And that's Chandler.

RACHEL : And that's Phoebe , and that's Joey.

RACHEL : And that's Phoebe , and that's Joey.

ROSS : Phoebe that's not true.

## 7.4 PCA4

### 7.4.1 Highest coefficient

ROSS : Well it's okay. Chandler is talking to her.

JOEY : I said a little bit Ross. Now, how about you Chandler?

JOEY : Okay. I'm Chandler

JOEY : Hey look Ross, you need to understand something okay? I uh...I am never gonna act on this Rachel thing, okay? I-I would never do anything to jeopardize my friendship with you.

JOEY : It's okay, Ross, alright? I totally understand. Of course you're not fine. You're.. You're Ross and Rachel.

JOEY : I'm fine, I'm fine, it's just, it's just weird what's happening with her and Ross. You know, yesterday he asked me to fix him up with somebody.

RACHEL : All right. So you're telling me that there is nothing going on between you and Chandler.

ROSS : Fine, fine, Rachel your with Monica, Joey you're with me.

PHOEBE : Okay. Oh umm, Chandler, Monica is looking for you.

ROSS : Umm, okay, yeah, sure. But wh-what's wrong with Monica and Chandler?

### 7.4.2 Lowest coefficient

MONICA : What?! What is it?!

MONICA : Oh my God! I love that!

JOEY : What the hell is that?!!

JOEY : What the hell!

ROSS : What?! It is?!

RACHEL : Oh my God! That's the creepiest thing I've ever heard!

RACHEL : Oh my God! Look at this!

MONICA : What?! What is it?

ROSS : I can't believe this!!

JOEY : What?! What?!! What is it?!

## 7.5 PCA5

### 7.5.1 Highest coefficient

RACHEL : Yeah! That would be great!

MONICA : Yeah, that'd be great! Thank you!

JOEY : Yeah! Yeah! That would be very helpful! Yeah.

CHANDLER : All right, ready?

ROSS : All right, ready?

CHANDLER : All right, ready?

PHOEBE : All right, ready?

MONICA : All right, you ready?

PHOEBE : Sure, yeah!

JOEY : Sure. Yep.

### 7.5.2 Lowest coefficient

PHOEBE : But, also, what happened between you and your Mom?

JOEY : She was nothing compared to you.

JOEY : She was nothing compared to you.

CHANDLER : Hey that's what I tell girls about me.

JOEY : Me too. I mean I...haven't thought at all about how I put myself out there and said all that stuff and how you didn't feel the same way about me and-and how it was really awkward.

ROSS : Well, well I am married. Even though I haven't spoken to my wife since the wedding.

PHOEBE : Oh, because, you know... they don't like you.

MONICA : Well, um, because mainly, um, they don't like you. I'm sorry.

CHANDLER : Well it couldn't have been worse. A woman literally passed through me. OK, so what is it, am I hideously unattractive?

ROSS : Hey, whatever it is, I am sure it has happened to me. Y'know, actually once-once I got dumped during sex.

## 7.6 PCA6

### 7.6.1 Highest coefficient

ROSS : Yes! We're getting married?!

JOEY : No! No, and I did not ask her to marry me!

ROSS : N-no! Okay? We've been through this! We're not gonna get married just because she's pregnant, okay?

JOEY : Well all right then, I guess I shouldn't get to excited about the fact that I just kissed her!

CHANDLER : OH...MY...GAWD! I am so sorry sweetie, are you okay? You didn't tell her we were getting married, did you?

ROSS : Hey! I offered to marry her!

CHANDLER : How can I not be upset? Okay? I finally fall in love with this fantastic woman and it turns out that she wanted you first!

PHOEBE : You're still gonna go out with her?!

ROSS : Yeah? Oh-oh, she'd be so excited!

ROSS : Okay. I did divert her and we ended up having a great time! Okay?

### 7.6.2 Lowest coefficient

PHOEBE : Wait a minute. What's his name?

MONICA : Hey. It's him. Who is it?

MONICA : Nothing, I don't know.

JOEY : Seriously, who is this guy?

JOEY : Who the hell is this guy?

RACHEL : Who are these men?

PHOEBE : Come on, give me something. What's his name?

CHANDLER : There's the man.

MONICA : Who, who are they?

ROSS : C'mon, what's his name?

## 7.7 PCA9

### 7.7.1 Highest coefficient

CHANDLER : What are you guys doing together?

RACHEL : So what are you guys going to do?

ROSS : What are you guys doing later?

MONICA : So, what have you guys been doing?

ROSS : Well, I'm gonna go see her. I want to bring her something, what do you think she'll like?

MONICA : What are you guys gonna do?

ROSS : So uh, any ideas for the bachelor party yet?

RACHEL : What're you guys doing out here?

ROSS : Hey, what have you guys been up to?

RACHEL : Hey, what have you guys been up to?

### 7.7.2 Lowest coefficient

PHOEBE : Oh, okay, oh.

ROSS : Oh. Oh! Oh my God! Okay, I know this, give me-give me a second!

PHOEBE : All right-Ooh! Oh dead God, save me!

RACHEL : Oh-oh, sorry, it's this way, it's this way.

RACHEL : Oh, okay!

CHANDLER : Oh, okay!

RACHEL : Oh, okay!

MONICA : Oh, okay!

ROSS : Oh, you're right, I'm sorry.

JOEY : Oh, oh, oh, sorry.

### 7.8 PCA18

#### 7.8.1 Highest coefficient

JOEY : Yeah, he did, look... look, it's right there on the counter! Ha-ho-ho!

CHANDLER : Okay, did you see that?! With the inappropriate and the pinching!!

CHANDLER : Okay, did you see that?! With the inappropriate and the pinching!!

JOEY : Hey! Handcuffs! And fur line, nice! I didn't know you guys had it in ya!

JOEY : Look, it was a job all right?

CHANDLER : Look! Look! Look what the... Look what... Look what the floating heads did!

ROSS : Okay, there was some staring and pointing.

MONICA : Yeah, yeah, it's interesting.. but y'know what? Just for fun, let's see what it looked like in the old spot. Alright, just to compare. Let's see. Well, it looks good there too. Let's just leave it there for a while.

JOEY : Uh, take a look at the guy's pants! I mean, I know you told us to show excitement, but don't you think he went a little overboard?

RACHEL : Yeah, he did! Oh, see, this is what I'm talking about!

#### 7.8.2 Lowest coefficient

RACHEL : Oh no.

PHOEBE : Oh no.

PHOEBE : Oh no.

RACHEL : Oh no.

CHANDLER : Oh no.

PHOEBE : Oh no.

ROSS : Oh no.

PHOEBE : Oh no.

PHOEBE : Oh no.

ROSS : Oh no.

### 7.9 PCA7

#### 7.9.1 Highest coefficient

CHANDLER : What? What?

CHANDLER : What? What?

ROSS : What? What?

ROSS : What? What?

PHOEBE : What? What?

ROSS : What? What?

JOEY : What? What?

ROSS : What? What?

ROSS : What? What?

MONICA : What?

### 7.9.2 Lowest coefficient

JOEY : Yeah, yeah, I met this woman.

MONICA : Yes but my mom got me this job.

PHOEBE : Yes, yes I do. God, oh it's just perfect! Wow! I bet it has a great story behind it too. Did they tell you anything? Like y'know where it was from or...

PHOEBE : No, not usually. But yeah, I could use one right now.

PHOEBE : Yeah, kinda.

MONICA : Yeah, just like the one in the poem.

CHANDLER : Yes, money well spent!

PHOEBE : No! But it's the nicest kitchen, the refrigerator told me to have a great day.

CHANDLER : Yeah, I remember.

MONICA : No. But I remember people telling me about it.

## 7.10  PCA17

### 7.10.1  Highest coefficient

RACHEL : And um, what-what is that Ross?

RACHEL : Ross's what?

RACHEL : Ok, Ross, Ross, ok listen, what we have is amazing.

CHANDLER : Oh, that's Ross's.

CHANDLER : Oh, that's Ross's.

RACHEL : Ross, I...

RACHEL : For Ross, Ross, Ross.

RACHEL : Well-well, I don't know Ross-really?

RACHEL : Well-well, I don't know Ross-really?

RACHEL : Um... Ross?

### 7.10.2  Lowest coefficient

MONICA : Hey, Joey, I don't think that you should leave Chandler alone. I mean it's only been two days since he broke up with Kathy. Maybe you can go fishing next week?

JOEY : Chandler, you have to start getting over her. All right, if you play, you get some fresh air, maybe it'll take your mind off Janice, and if you don't play, everyone will be mad at you 'cause the teams won't be even. Come on.

PHOEBE : Joey? How could you just let them leave?

CHANDLER : Look, Joey, Kathy is clearly not fulfilling your emotional needs. But Casey, I mean granted I only saw the back of her head, but I got this sense that she's-she's smart, and funny, and gets you.

MONICA : Wait a minute...Joey. Joey you can't ask her out, she's your roommate. It-it'll be way too complicated.

PHOEBE : Okay, but try and get Joey too.

ROSS : No Joey! Look why don't, why don't we just let her decide? Okay? Hey-hey, we'll each go out with her one more time. And-and we'll see who she likes best.

RACHEL : Yeah, Joey kinda disabled it when I moved in.

1105 MONICA : Joey that is horriable.

1106 CHANDLER : No, see the thing is I want to get out of here before Joey gets all worked
1107       up and starts calling everybody bitch.

## 7.11 PCA16

### 7.11.1 Highest coefficient

1110 RACHEL : Oh, oh. . What is this?

1111 PHOEBE : Oh, yeah. What's this?

1112 JOEY : I don't know. It's-it's just...lately, I've been feeling... Okay, here's what it is...
1113       You know what? I feel a lot better, thanks!

1114 PHOEBE : Ohh. What is this?

1115 CHANDLER : Oh-oh, what are you doing?

1116 PHOEBE : Oh that's so great! Ohh, so what's going on now?

1117 PHOEBE : Oh my God, what's it doing here?

1118 JOEY : Yeah! Yeah, why? What's up?

1119 PHOEBE : Oh, why? What's up?

1120 PHOEBE : What-what's up?

### 7.11.2 Lowest coefficient

1122 CHANDLER : And then he did.

1123 PHOEBE : And we did.

1124 ROSS : No you didn't. You said you would, but you never did!

1125 CHANDLER : I sure did.

1126 RACHEL : No, you could've lost your job.

1127 ROSS : Sure, Monica would have to give her up.

1128 CHANDLER : Yes he did.

1129 RACHEL : That is not true. She did! She forced me!

1130 RACHEL : That is not true. She did! She forced me!

1131 ROSS : Monica! Would it?

