# OpenReview forum: "A test of stochastic parroting in a generalisation task: predicting the characters in TV series"
_NeurIPS.cc/2024/Conference — Submitted to NeurIPS 2024_

### Official Review · Reviewer_3PCU · 2024-07-07

**Soundness:** 1
**Presentation:** 2
**Contribution:** 1
**Rating:** 3
**Confidence:** 4

**Summary:**

The authors present an analysis of logistic regression on sentence embeddings as a way to predict the speaker of a particular line of dialogue in the Big Bang Theory. Specifically, they fit a PCA model to embeddings obtained from a sentence transformer and then use each PCA dimension as a linear feature. The authors present some qualitative analysis of the most predictive PCA dimensions and some quantitative analysis of the classification accuracy. In addition, they present a brief analysis of the ability of GPT-4 to directly classify lines of dialogue and compare to a limited user study.

**Strengths:**

The authors have identified an interesting and important debate in the AI community — more tests which help researchers discriminate between mere stochastic parroting and true generalization are certainly needed! In addition, the authors are very thorough in their description of the methods involved and their qualitative analysis of the PCA features is extensive. I also appreciate the thought the authors have given to the limitations of their study and the need for further work.

**Weaknesses:**

First and foremost, I feel that this paper needs to be much clearer and more focused in its research question. The introduction indicates that the objective of the study is to determine the extent to which the apparent ability of large language models like GPT-4 to generalize to novel tasks is actually attributable to their ability to parrot data from their training. However, a good part of the analysis appears dedicated to the specifics of The Big Bang Theory and the features of its dialogue. Section 3.1, for instance, extensively interrogates the PCA dimensions obtained from the sentence embeddings in a way that feels very specific to the particular dataset. Similarly, the conclusion raises claims that the ability for logistic regression to predict the speaker with reasonable accuracy is due to stereotyping in the characterization of the show. These claims are potentially warranted given the experimental evidence (though a more detailed and statistically-motivated analysis would be necessary to make such claims with certainty), but feel as though they belong in a different paper (a potentially quite interesting paper for a different venue, I should add). The connection between these results and the initial framing of LLM evaluation remain, unfortunately, somewhat murky. This is not to say there is no possible link between dialogue speaker prediction and LLM abilities! I encourage the authors to think about this problem more and articulate the specific claim they hope to interrogate.

On that note, and assuming that the main motivation is indeed to study large language models, I feel that the analysis could be strengthened. First, it would be helpful to justify some of the specific decisions made as part of evaluation. For instance, why were the Big Bang Theory and Friends selected over other possible dialogue datasets? Why was dialogue speaker prediction always studied between exactly two characters? Why were these specific characters selected? Do the characters have a similar amount of lines, or are there other statistical biases in the dataset that might affect the results?  When proposing a novel task, it’s important to make the assumptions and decisions that went into the task selection clear.

With regards to human evaluation, I encourage the authors to widen their study. That is to say, a user study which consists of only two participants (both of whom are related to one of the authors) makes it difficult to ascertain the reliability of the results. Indeed, I would suggest a study consisting of a larger number participants (ideally participants who do not have any externally motivating factors like relationships to the authors) so that a more general measure of human ability can be obtained. Further, I think it could actually be preferable for the participants to not have prior experience with the television show. This would make the test more an examination of the ability for participants to generalize their knowledge of personality traits to a novel situation instead of their ability to recall information (which is, ostensibly, closer to the desired research question in LLMs).

Despite these critiques, I hope that the authors continue to refine their research question, justification, and methodology. There are interesting questions to study here!

**Questions:**

**Questions**
- See above for questions on task selection
- How does this classification task differ from those previously studied? e.g. those used in https://arxiv.org/abs/2309.07755 or https://arxiv.org/abs/2402.07470
- Is there any effect of the prompt on the downstream results? (For GPT-4)
- Is there any effect of random seed on the downstream results? (All models)

**Notes**
- The paper could benefit from an additional round of proofreading (e.g. “analyze” in Section 2.2 —> “analysis”, “annex” —> “appendix” throughout the paper)
- The paper contains links to a few public GitHub repositories, which technically may violate double-blind review. The authors should be careful about including identifiable information in anonymous submissions

**Limitations:**

I feel that the authors have been very up front with the limitations of their work and have situated it in the context of broader impacts.

---

### Official Review · Reviewer_c8T5 · 2024-07-11

**Soundness:** 1
**Presentation:** 3
**Contribution:** 1
**Rating:** 2
**Confidence:** 5

**Summary:**

The authors are focused on whether or not LLMs can be thought of as stochastic parrots or contain "Sparks of AGI". They look into what kind of data is recoverable from internal LLM representations. Specifically, the authors investigate to what extent the task of identifying TV personalities (e.g. Penny vs Sheldon) based on their dialogue lines is solvable using various methods. The authors compare a classifier based on PCA components extracted from existing LLM embeddings, GPT-4 zero shot performance, and human expert judgments. They find that all methods show fairly good performance, with human experts showing best results, followed by GPT-4, followed by the classifier. The authors also present a brief qualitative analysis, interpreting the more prevalent axes of variation in the embeddings identified using PCA.

**Strengths:**

The authors tackle a very ambitious and important problem. The writing is clear throughout, and the authors provide extensive background for the methods they use.

**Weaknesses:**

I need to preface this by saying that I hope that my negative review does not discourage the authors from further pursuing the topic. I feel bad for having to reject this paper as it has some good ideas behind it and has an intention of researching a highly important problem. I hope that in next iterations, their work can be improved and expanded. At present, unfortunately, it does not match publication standards. I will try to explain why, and give pointers on how to potentially fix it in the future.

The biggest flaw of the paper is the experiment design. The authors never clearly define what exactly it means to be a "stochastic parrot" as opposed to "general intelligence". The authors also don't explain how their experiments would help to decide one way or another. So the results we have are impossible to interpret. It would help to go back to the original question and work through the argumentation more clearly. If the internal LLM representations have information about TV personalities, does it make them more or less of a stochastic parrot and why.

Otherwise, the experiments give a very exploratory impression. For example the authors run PCA on sentence embeddings computed on their dataset and interpret the components. But it's unclear why and how this would help to answer the main question the paper attempts to answer.

Additionally, the paper's methods are extremely well-known, but unfortunately, the authors don't refer to relevant literature. The work highly overlaps with the topic of linear and nonlinear probes, as well as with the general theme of transfer learning. In essence, what the authors did can be described as adding a "classification head" to a pre-existing LLM. This is a very well-known technique.

If we want to gain new insights into what the models are doing, it is usually more interesting to look into the computations in intermediate layers of the model, rather than the last embedding layer. It is also often desirable to look at causal probes (rather than just a classifier).

Lastly, there are certain writing choices that deviate from common "conventions" in academic publishing. For example, oftentimes the authors go into excessive detail on well-known methods (explaining how PCA works and what a covariance matrix is). I highly suggest that the authors look at existing successful papers that use similar methods and copy their approach when it comes to decisions on what to explain in the main text, what to put into the appendix, and what to omit. The general rule of thumb is that newly introduced and important ideas should be at least briefly given in the main text, with extra details given in the appendix. Extremely well-known and established methods such as Principal Component Analysis don't need a full explanation, and a simple reference to the original source is enough.

I really hope that the authors don't get discouraged and try to refine and improve their research in the future. The first starting point would be to more clearly define the problem, and to study in depth the existing literature on linear probes and probing in general, and on investigating what the internal LLM representations contain. One potential starting point is the paper "Evaluating the World Model Implicit in a Generative Model", Vafa et al. 2024 and related works.

**Questions:**

How exactly do we go from studying internal representations to the conclusions about stochastic parroting? For example, there are Neuroscience works that show that one can recover a lot of what a person is thinking about (see the studies related to the "Grandmother cell" idea). A lot of visual input can be reconstructed as well. Does it make humans stochastic parrots as well?

Basically, if GPT-4 can answer a given question (Penny vs Scheldon), we already know that its internal representations contain information needed to answer that. Same goes for human brains. I don't fully understand how it relates to the question of "stochastic parrotedness".

**Limitations:**

The authors acknowledge some of the limitations of their study.

---

### Official Review · Reviewer_rrdr · 2024-07-13

**Soundness:** 1
**Presentation:** 1
**Contribution:** 1
**Rating:** 1
**Confidence:** 5

**Summary:**

This paper’s main contribution is to apply a logistic regression on the principal components of the LLM embeddings for classifying TV series characters based on their dialog lines. The main finding is the logistic regression approach does worse than GPT-4 in predicting TA characters, but is comparable to human evaluations with two annotators.

**Strengths:**

The paper focus on an interesting angle of using language model features for predicting the belongings of dialog lines of characters of TV shows.

**Weaknesses:**

The methodology of using logistic regression over PCA of language model embeddings is not novel, and there's no rigorous quantitative evaluations of the method beyond qualitative examples. The connection of the method and task to the broad discussion around "spark of AGI" and "Stochastic Parrots" is farfetched.

**Questions:**

Could you address the technical novelty of the method proposed?

**Limitations:**

The paper claims that "the contribution of the paper is primarily methodological, and their study is limited to a qualitative study of two very specific datasets." However, the method they adopt is a fundamental ML technique, which lacks novelty.

---

### Official Review · Reviewer_ynzH · 2024-07-19

**Soundness:** 2
**Presentation:** 3
**Contribution:** 1
**Rating:** 3
**Confidence:** 4

**Summary:**

The paper aims to prove LLMs work as "stochastic parrots" (Bender et al) rather than "sparks of agi" (Bubeck et al). To prove this claim, the paper presents an experiment where a task can be solved by training a linear model (logistic regression) on top of PCA of the LLM output. The authors then claim, based on the linear model experiments, that the LLM doesn't exhibit any sparks of agi due to the ability of (nearly) solving the task using linear models.

**Strengths:**

The authors show a simple linear model trained on the output of an LLM for a given task is good enough to solve it, compared to using a GPT4 model, raising questions on the supposed intelligence often ascribed to the model.

**Weaknesses:**

While I generally agree that LLM's are closer to "stochastic parrots" than "sparks of agi", the claim that it can be proved using the proposed PCA experiments is weak to me.

- Firstly, the embeddings are essentially the output of the LLM in question (all-MiniLM-L6-v2) - I would call it outputs rather than embeddings, as embeddings just indicate input word embeddings to the model, which clearly here isn't the case.
- Secondly, the outputs itself being feature rich to be used for classification is unsurprising. It is expected the principal components of this embedding would be useful in predicting the properties of the task (as shown in the projection of PCA plots). This just shows the underlying model (SentenceBERT here) is good at extracting rich semantic and syntactic features from the input sentence (probing literature essentially proves that [1]).
- Lastly, the experiment also shows the representations extracted from the sentence embedding model is sufficient for the task. For a harder task, if the linear probe on all-MiniLM-L6-v2 was not good with respect to GPT4, that would also not conclusively prove the ability of GPT4 is due to any sparks, rather it can be explained that GPT4's own embedding features are richer. That is, a linear probe trained on GPT4 embeddings for a harder task would also likely mimic its own performance. (this is theoretical, as neither the author or anyone other than OpenAI have access to their embeddings)

[1] https://aclanthology.org/D19-1250/

**Questions:**

None

**Limitations:**

There are no explicit limitation section, however the last paragraph of conclusion discusses it.

---

### Decision · Program_Chairs · 2024-09-25

**Decision:**

Reject

**Comment:**

This paper experiments with training a linear model on LLM's internal representations of inputs to extract semantics encoded in the LM parameters, with the goal of addressing the broad philosophical question of whether LLMs are "stochastic parrots" or genuinely "generally intelligent". There is not sufficient technical novelty compared to prior work, which isn't referenced. I echo the sentiment of the review by c8T5 in that I hope the authors will continue to refine their work and are not discouraged (there are great suggestions from the reviewers).